# Monitoring methaemoglobinaemia in birds using 5 µL of whole blood

Clive A. Marks[1,2]*, Katherine Trought[2], Samantha Brown[2], Jane Arrow[2], Brian Hopkins[2]

1 Nocturnal Wildlife Research Pty Ltd, Melbourne, Australia, 2 Manaaki Whenua Landcare Research, Lincoln, New Zealand

* clivemarks@NocturnalWR.com.au

**Data Availability Statement:** Data are available on Figshare (DOI: 10.6084/m9.figshare.22188127).

**Funding:** This work was funded by the Department of Conservation (New Zealand) Predator Free 2050 Tools to Market Programme under contract

## Abstract

Methaemoglobin (MetHb) forming compounds such as para-aminopropiophenone (PAPP) and sodium nitrite ($NaNO_2$) have recently been adopted for the lethal control of a range of invasive carnivores and mustelids. Determining the relative hazard of these compounds to non-target bird species is an important component of ecological risks evaluation. Problematically, some potential non-target bird species may be as small as 10 g in body mass, thus placing limitations on blood volumes that can be routinely sampled. Accordingly, we developed methods to quantify markers of increasing methaemoglobinaemia at their point of collection that required only 5 µL of whole blood. A 3 µL blood aliquot is pipetted into a plastic micro-cuvette and placed in a custom made holder optically coupled to the Ocean Optics spectrometer, enabling absorbance for oxyhaemoglobin (HbO: 575 nm) and MetHb (630 nm) to be determined. Haemoglobin ($HbFe^{2+}$), packed cell volume (PCV) and lactate (LAC) data were generated from the remaining 2 µL aliquot apportioned to biosensor strips for the Cera-Check® and Lactate Scout® point-of-care devices. After oral doses of PAPP, a methaemoglobinaemia absorbance index (MAI = absorbance 575 nm–absorbance 630 nm) was strongly and significantly associated with dose-dependent declines in $HbFe^{2+}$ in 9 bird species. Quantifying dose-dependent responses to MetHb-forming agents at the point of sample collection avoids analytical and storage artifacts arising from sample degradation that appears to be a much greater problem in avian blood compared to mammalian blood.

## Introduction

### Methaemoglobin-forming compounds

Haemoglobin (Hb) is normally maintained in the ferrous valence ($HbFe^{2+}$) enabling it to form oxyhaemoglobin (HbO) for the transport of oxygen ($O_2$) used in aerobic metabolism, as well as carbaminohemoglobin that contributes to the removal of its by-product, carbon dioxide ($CO_2$). When oxidised to the ferric state ($HbFe^{3+}$), known as methaemoglobin (MetHb), Hb is unable to bind to $O_2$ as its sixth coordination site becomes occupied by a water molecule or hydroxyl group [1]. Incomplete oxidation of any of the four Hb subunits (two $\alpha_2$ and two $\beta_2$ chains) will impart an increased affinity for $O_2$ that remains bound to any of the sub-units [2]

(Reference Number: 3054988; Contract Report: LC4147) and a research programme grant (2223-28-009 A) funded by the Ministry of Business, Innovation and Employment (Precision Pest Eradication - pest-selective control tools, C09X2208). The funders had no role in study design, data collection and analysis, decision to publish, or preparation of the manuscript. Staff of the Department of Conservation assisted in the coordination of work that was part of ongoing species recovery programmes.

**Competing interests:** The authors have declared that no competing interests exist.

causing a shift in the oxygen dissociation curve to the left and preventing its release [3]. Due to autoxidation alone, some 1% of the blood's total pool of Hb may exist as MetHb at any time [4], yet the presence of MetHb does not become pathological unless at concentrations sufficient to inhibit aerobic metabolism. MetHb-forming compounds may cause a deepening hypoxia associated with ever greater inhibition of aerobic ATP production and increased generation of reactive oxygen species [ROS] [5]. If severe, the inhibition of aerobic metabolism can result in a loss of ion homeostasis and irreversible cellular damage [6] in vital organs such as the brain and heart that can be fatal [7].

Para-aminopropiophenone (PAPP) is a potent MetHb-forming compound [8] used as a lethal control agent for invasive carnivores and mustelids in New Zealand and Australia [9–11]. Sodium nitrite ($NaNO_2$) is another MetHb-forming agent registered for the control of brushtail possums in New Zealand [12] and feral pigs in Australia [13]. In mammals, rapid and sustained elevation in MetHb can quickly cause unconsciousness prior to death, a characteristic that makes MetHb-forming agents potentially more humane than other conventional pest control agents [10]. Establishing the hazard of MetHb-forming compounds to non-target bird species is an important component of risk management and regulatory toxicology. Lethal-dose bioassays conducted in the early 1940s [14] implied that carnivores and mustelids were more acutely sensitive to PAPP than birds [15]. However, as the health and ecological fitness of birds is also highly dependent on maintaining haemoglobin in the ferrous valance ($HbFe^{2+}$) able to transport $O_2$ to meet metabolic and functional demands [16], assessing the impact of PAPP on $HbFe^{2+}$ concentrations and HbO-mediated $O_2$ transport in avifauna is a key marker of their relative sensitivity to MetHb-forming agents. As an alternative to bioassays that use dichotomous lethal-dose data (survival or death) to establish the hazard of MetHb-forming agents, we sought continuous data to directly measure the dose-dependent impacts caused by MetHb-forming agents. This approach was selected to reduce animal use and to permit dose-response modeling that avoids the need to use death as an experimental endpoint in high conservation value bird species [17].

## The blood sampling problem in birds

Typically, between 200–300 μL of blood is required to make a comprehensive haematology examination [18], although such blood volumes cannot be safely taken from birds with a small body mass (BM). Of some 6000 extant bird species, their median BM was reported to be 37.6 g, with a frequency distribution skewed below this value [19]. Low BM in the class *Aves* has practical implications for the maximum blood volumes that can be harvested from most birds, as the removal of significant volumes of blood may compromise fitness and confound experimental dose-response data that rely upon unperturbed haematological profiles. For example, blood sampling of cliff swallows (*Petrochelidon pyrrhonota*) was associated with a 21–33% reduction in onward bird survival that was attributed to hypovolemic shock [20]. Present day guidelines recommend that blood volumes taken from birds should be restricted to no more than 1% of their BM [21] which represents approximately 10% of their total blood volume [18]. When such guidelines are followed there appears to be little evidence that blood sampling negatively impacts upon wild bird survival [22]. Nevertheless, when monitoring blood values to investigate changing physiological states, a number of sampling events may be required over several hours, or days. Here, blood wastage due to syringe use may also become a far more significant component of the overall sample volume in low BW birds, as up to 10 μL of blood may remain in the needle body and hub of some syringes [23]. Over 6 sampling events in a 10 g bird (10 g x 0.01 = 100 μL blood volume limit), over half of the sample (60 μL) is likely to be unavailable for analysis.

CO-Oximeters used for laboratory or point-of-care analysis to determine the concentration of different haemoglobin species typically require minimum sample volumes ranging from 50 μL [24] to 300 μL [25]. While blood samples as small as 45 μL have been lysed, centrifuged and added to a phosphate buffer to permit MetHb analysis in the laboratory [26], several factors suggest that rapid analysis of avian blood at the point of collection should be preferred. Avian blood is difficult to preserve [27], moreover, changes in MetHb and HbO in response to MetHb-forming agents can be highly dynamic [10, 28] and may alter due to changes in the induction of MetHb-reductases [29]. Accordingly, in order to facilitate investigations of the impact of MetHb-forming agents on a wide range of birds, we sought to develop methods able to analyse sequential whole blood samples no more than 5 μL in volume at their point of collection to provide markers of comparative hypoxaemia and hypoxia.

## Materials and methods

### Ethics statement

All procedures were compliant with the New Zealand *Animal Welfare Act* (1999) and the 2019 *Code of Ethical Conduct for the Use of Animals for Research, Testing and Teaching* published by the New Zealand Department of Conservation (DOC). Experimental procedures were approved by the Manaaki Whenua Landcare Research Animal Ethics Committee as protocol 17/12/01. Use of bird species and access to study sites was approved and licensed by DOC and the relevant species recovery groups. The capture of birds was covered under the DOC Global Concession Permit (CA-31615-OTH).

### Instruments

We assessed a wide range of point-of-care devices and customised sensors to assess haemoglobin oxygen status yet report only on preliminary *in vitro* trials and final *in vivo* validation of the Cera-Chek® Hb point of care monitor (Ceragem Medisys: Chungnam, South Korea) used in combination with a customised absorbance spectrometer. *In vitro* comparisons were initially made against the Avoximeter 4000® CO-Oximeter (Werfen, Barcelona, Spain) that used a 50 μL cuvette. For all *in vivo* bird trials, we used the Flame® absorbance spectrometer with a sensor bandwidth of 350–1000 nm (Ocean Optics: Largo, USA) with a HL-2000 Vis-NIR spectrometer light source (Ocean Optics: Largo, USA) that has a spectral range of 360–2400 nm set up and calibrated using the manufacturer's instructions for dark-field and light-field calibration with blank cuvettes. Absorbance data were generated over the full range of the spectrometer (350–1000 nm) from 3 μL blood samples pipetted into a disposable cuvette normally used by the DiaSpect® Tm Hb analyser (EKF Diagnostic: Cardiff, United Kingdom). The cuvette window was placed in the spectrometer's light path using a custom-built holder that was 3D printed from opaque black polylactic acid (PLA) and inserted into a machined aluminium block housing that accepted the SMA 905 connectors from the optical fibre terminators (Ocean Optics: Largo, USA). Lactate (LAC) was measured using the Lactate Scout® (EKF diagnostics, Germany) by pipetting between 0.5–0.8 μL of whole blood onto the biosensor and reading the result as mmol $L^{-1}$. A further 1 μL of whole blood was pipetted to the biosensor of the Cera-Chek total haemoglobin (Hbt) and packed cell volume (PCV) point-of-care monitor and read as g Hb $dL^{-1}$ and % of the cell fraction in the sample respectively (Fig 1).

### Preliminary *in vitro* evaluation of instruments

The Cera-Check device was first assessed in preliminary *in vitro* trials using varying concentrations of MetHb produced by a gradient of sodium nitrite ($NaNO_2$). Unlike PAPP, $NaNO_2$

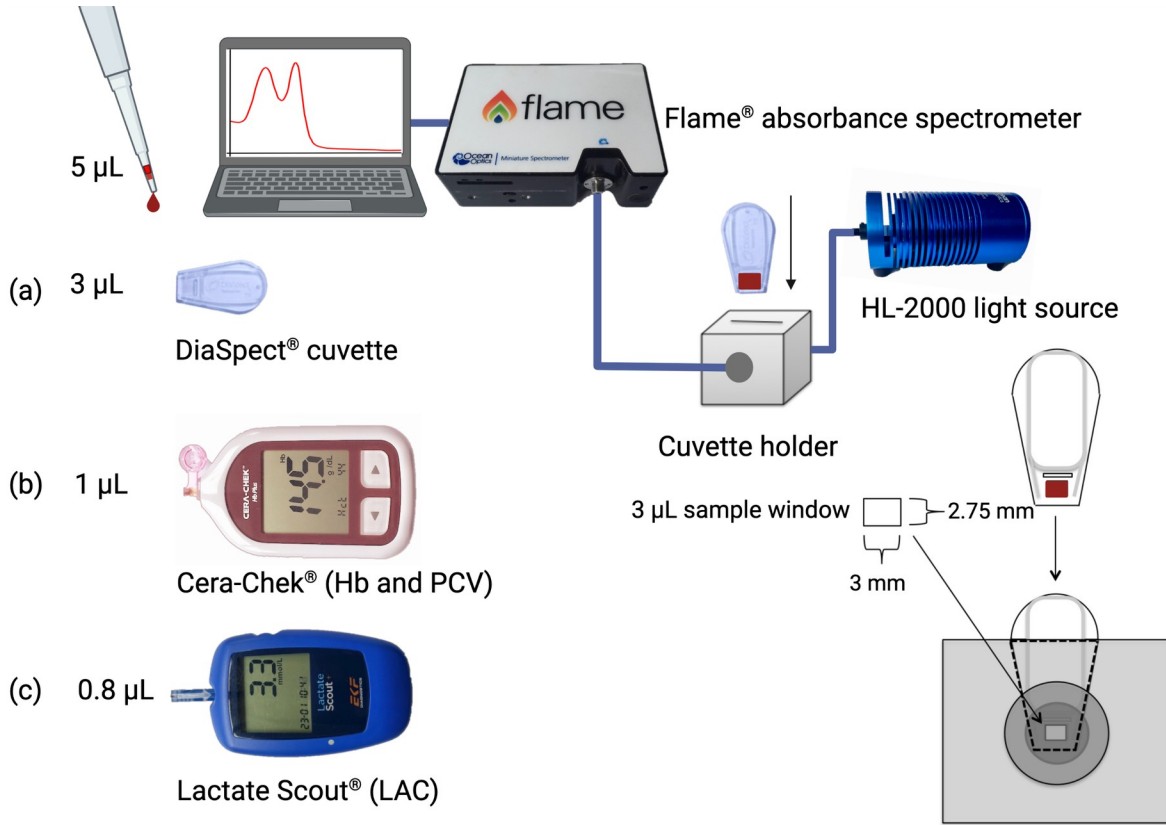

**Fig 1.** A 5 μL whole blood sample was apportioned to (a) the DiaSpect plastic cuvette (3 μL) and placed in a customised holder with SMA 905 connectors for the optical fibre terminators for the Flame absorbance spectrometer (Ocean Optics: Largo, USA). Another 1 μL of whole blood (b) was pipetted to the biosensor of the Cera-Chek total haemoglobin (Hbt) and packed cell volume (PCV) point-of-care monitor. Lactate (LAC) was measured using the (c) Lactate Scout (EKF diagnostics, Germany) by pipetting between 0.5–0.8 μL of whole blood onto the biosensor.

does not require biotransformation to an active to produce MetHb and directly oxidises Hb [30]. We chose chickens (*Gallus gallus*) blood as a bird model and three large mammal species from which at least 30 mL of blood could be harvested. Procedures followed those described for human blood (Shihana *et al.* 2010) and data obtained in carnivores using cat (*Felis catus*) and dog (*Canis domesticus*) blood, although Patton et al. (2016) had used coyote (*Canis latrans*) blood. Fresh blood was taken from brushtail possums (*Trichosurus vulpecula*) located at the animal house (Lincoln) and stored in lithium heparin vacutainers. Blood from cats, dogs and chickens were accessed from the New Zealand Companion Animal Blood Bank (Palmerston North, New Zealand) where samples were collected in 30 mL syringes and also preserved in lithium heparin. Blood was transported and stored by chilling to 6˚C and used the following day when 0.9 mL of blood was decanted into 1.7 mL microcentrifuge tubes and placed in a dry-bath incubator (MiniB-100, Hangzhou Miu Instruments: Zhejiang, China) set to the core body temperature for the appropriate animal (possum = 36.5˚C; chicken = 41.0˚C; dog = 36.5˚C; cat = 38.4˚C) and allowed to equilibrate for an hour. A dilution series of sodium nitrite (NaNO$_2$) was created in PBS (10 mM phosphate, 150 mM sodium chloride) to produce a concentration gradient of 2.5, 5, 10, 15, 20, 40, 60 mmol L$^{-1}$ with a corresponding control (PBS only). A total of 100 μL of each sodium nitrite dilution (or control) was added to 900 μL of blood and incubated for a further 20 minutes while data were taken after 10 minutes when and asymptote for MetHb values had occurred. For 1-day old blood samples the relationship

between MetHb (%) as determined by the Avoximeter 4000 and HbtFe$^{2+}$ as determined from the Cera-Chek was contrasted along with the strength of the association between MetHb and HbO from the Avoximeter 4000. We routinely attempted to assess presence of hemolysis by a combination of visual inspection against a white background, light field microscopic examination and by comparing changes in absorbance at 414 nm [31].

## Dosing of birds with PAPP

Domestic chicken pullets (Hyline strain), pekin ducks (*Anas platyrhynchos*) and Japanese quail (*Coturnix japonica*) were obtained from a commercial supplier. Black-backed gulls (*Larus dominicanus*) and pukekos (*Porphyrio melanotus*) were captured from wild populations. Takahē were held at the Burwood Takahē Breeding Centre and North Island brown kiwi (*Apteryx mantelli*) and weka were held at the Ōtorohanga Kiwi House. Due to difficulties in accessing captive keas (*Nestor notabilis)*, we used wild caught eastern rosellas (*Platycercus eximius)* as a surrogate species. We developed bird handling, PAPP dosage and blood sampling procedures to minimize distress. Preliminary trials were conducted in domestic bird species first in order to reduce blood volumes and to refine methods that required the smallest number of repeat sampling. Consultation with wildlife veterinarians sought to identify ways to accommodate unique behavioral features encountered in new species so to reduce unique causes of stress. Details of the specific anaesthesia, handling procedures and dose-response data are presented elsewhere [17] and are only briefly restated here. A stock solution of 375 mg mL$^{-1}$ PAPP dissolved in dimethyl sulfoxide (DMSO) was prepared by heating the solution to between 38-41˚C with sonication. As different bird species varied greatly in their sensitivity to PAPP, a progression of standard oral doses typically began at 4.4 mg kg$^{-1}$ and were increased by a factor of x 1.5 in each consecutive bird until the signs of clinical methaemoglobinaemia were observed (i.e. cyanosis of mucous membranes, ataxia and unconsciousness) together with significant perturbations in the blood values. In highly tolerant birds, doses were increased by more than a factor of x 1.5 until perturbations in absorbance values were detected, enabling a series of doses to be selected to produce more intense clinical signs of methaemoglobinaemia. The overall range of PAPP doses used varied between 0.6–855 mg kg$^{-1}$ and the full range of doses used in each species are reported elsewhere [17]. All birds were lightly anaesthetised to allow baseline blood samples to be taken at T0, other than for kiwi and weka where no anaesthesia was recommended by attending veterinarians. In other birds an intramuscular (IM) injection of Zoletil (tiletamine and zolazepam) was administered into the breast muscle using a 29 G needle at 20 mg kg$^{-1}$ except for takahē which received a midazolam and butorphanol combination at 1.5 mg kg$^{-1}$ and 2 mg kg$^{-1}$ respectively using an IM injection with a 23 G needle, followed by administration of the GABA receptor antagonist flumazenil [32] at 0.1 mg kg$^{-1}$ at the end of procedure to antagonise the action of the benzodiazepine (midazolam).

## Blood sampling

After baseline blood samples were taken at T0 prior to oral dosing with PAPP, blood sampling was repeated at 1-hour and 3-hours post oral dosing. Venepuncture methods followed those published for larger birds (Pollock 2015) approved by the Wildlife Ethics Committee for the Department of Environment, Water and Natural Resources (South Australia), the Food and Agriculture Association (World Health Organisation), or those used during routine veterinary procedures for various native bird species. In general, birds were restrained and/or under light anaesthesia one axilla ('wingpit') of each bird had either some feathers cut to access a patch of bare skin or feathers were dampened with ethanol to sterilise the area and assist the visibility

and access to a vein. During blood sampling restrained birds were held on their side by a seated technician and blood sampled by another. In chickens and ducks, blood was typically sought from the brachial vein using a 23 G needle and syringe, in smaller birds' blood collection involved the use of a heparinised capillary straw after lancing the vein with a 25 G needle. Takahē, kiwi and weka were blood sampled from the medial metatarsal vein using a 25 G needle and 1 mL syringe.

### *In vivo* validation of the methaemoglobinaemia absorbance index (MAI)

Absorbance wavelengths discriminating MetHb, HbO and deoxyhaemoglobin (deoxy-Hb) were selected from published reference data [33]. A methaemoglobinaemia absorbance index (MAI) was determined spectroscopically from a 3 μL whole blood sample by calculating the difference between absorbance value at 575nm (HbO) and 630 nm [MetHb] where wavelengths were selected in preliminary trials in ducks that produced the most lineal relationship and latter confirmed in 9 bird species (Fig 2).

$$MAI = (Absorbance_{575nm} - Absorbance_{630nm})$$

To ensure that responses in the MAI to PAPP in all bird species closely followed fluctuations in the concentration of Hb in the ferrous $HbFe^{2+}$ state due to its oxidation to MetHb

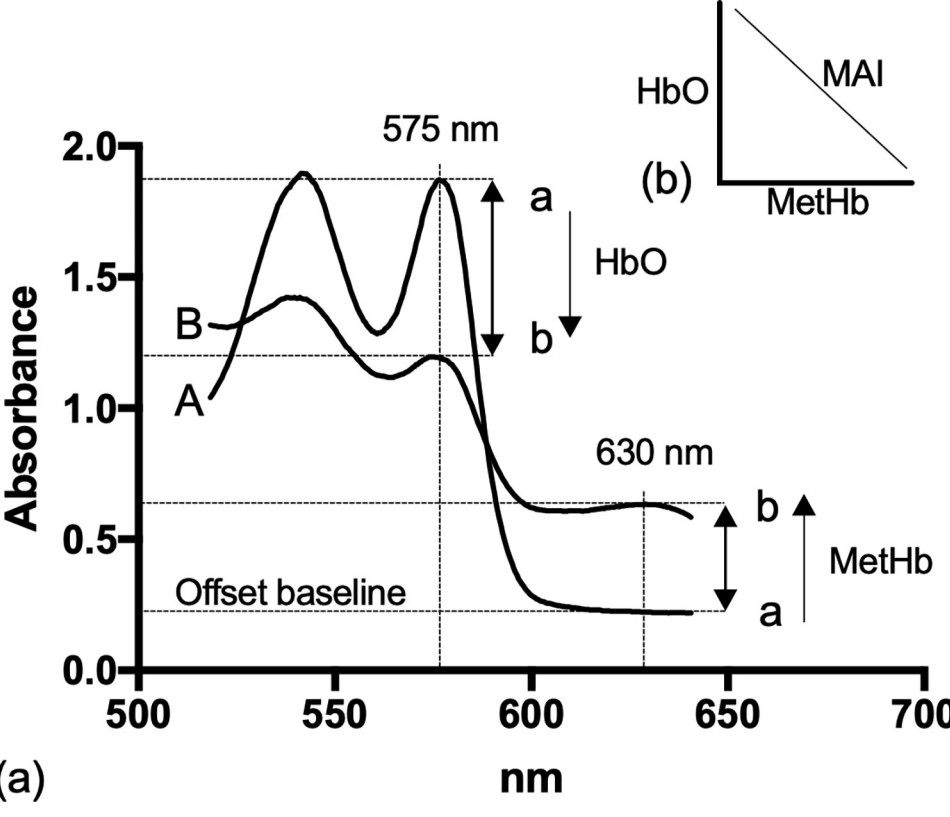

**Fig 2.** (a) Absorbance spectra of duck blood (A) prior to an oral dose of PAPP and (B) 180 minutes after receiving 74.2 mg kg$^{-1}$ of PAPP. The methaemoglobinaemia absorbance index (MAI) was based on dose-dependent increases in MetHb [a→b] measured at 630 nm that was inversely correlated with dose-dependent declines in HbO measured at 575 nm [a→b]. (b) The MAI was selected to be a highly linear continuous variable describing the severity of methaemoglobinaemia.

($HbFe^{3+}$), all MAI data for each bird species were regressed against $HbtFe^{2+}$ data from the Cera-Chek biosensor.

## Results

### Preliminary evaluation

After visual, microscope and spectroscopic examination at 414 nm, no indications of erythrocyte hemolysis were observed at any of the $NaNO_2$ concentrations. Preliminary *in vitro* evaluations of the Cera-Check biosensor showed that $HbFe^{2+}$ values declined in response to a gradient of $NaNO_2$ and were highly correlated with MetHb % determined by the CO-Oximeter for chickens ($R^2$ = -0.99, P < 0.0001), possums ($R^2$ = -0.97, P < 0.0001), cats ($R^2$ = -0.98, P < 0.0001) and dogs [$R^2$ = - 1.0, P < 0.0001] (Fig 3).

CO-Oximeter analysis revealed strong inverse correlations between MetHb and HbO in possums ($R^2$ = -1.0, P < 0.0001), cats ($R^2$ = -0.97, P < 0.0001), dogs ($R^2$ = -0.98, P < 0.0001) but not chicken blood [$R^2$ = 0.45, P = 0.1] (Fig 4).

Absorbance spectroscopy revealed that day-old chicken blood used for the *in vitro* comparisons had lost HbO absorbance peaks at 540 nm and 575 nm observed in freshly drawn blood and had developed dominant peaks at 553 nm corresponding to deoxyHb. Overall, large

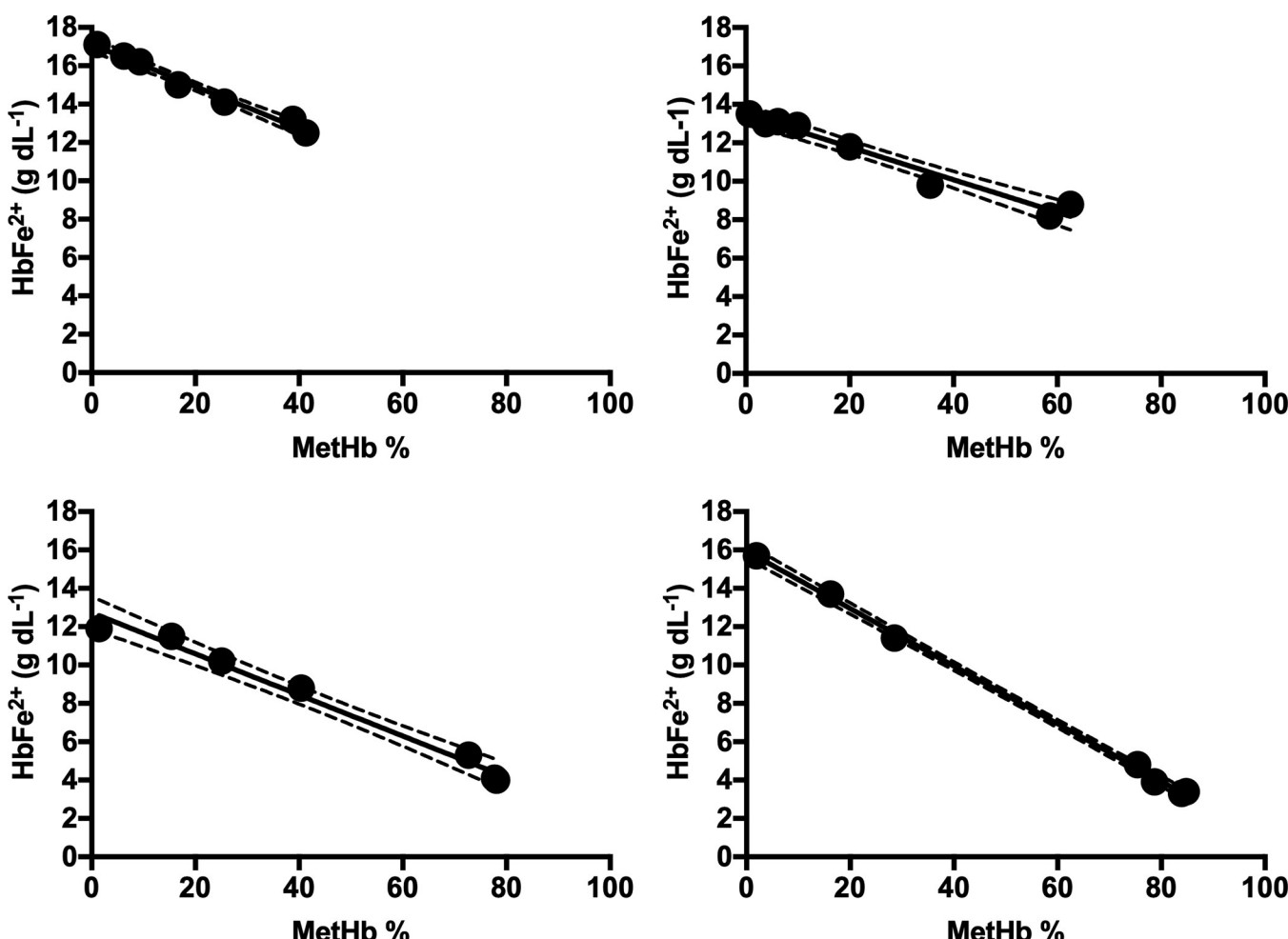

**Fig 3.** Haemoglobin ($HbFe^{2+}$ g $dL^{-1}$) derived from Cera-Chek regressed with methaemoglobin (MetHb %) determined by the Avoximeter 4000 CO-Oximeter using whole blood aliquots incubated with an *in vitro* gradient of 0, 2.5, 5, 10, 15, 20, 40 mmol $L^{-1}$ $NaNO^2$ for (a) chicken, (b) possum, (c) cat and (d) dog blood.

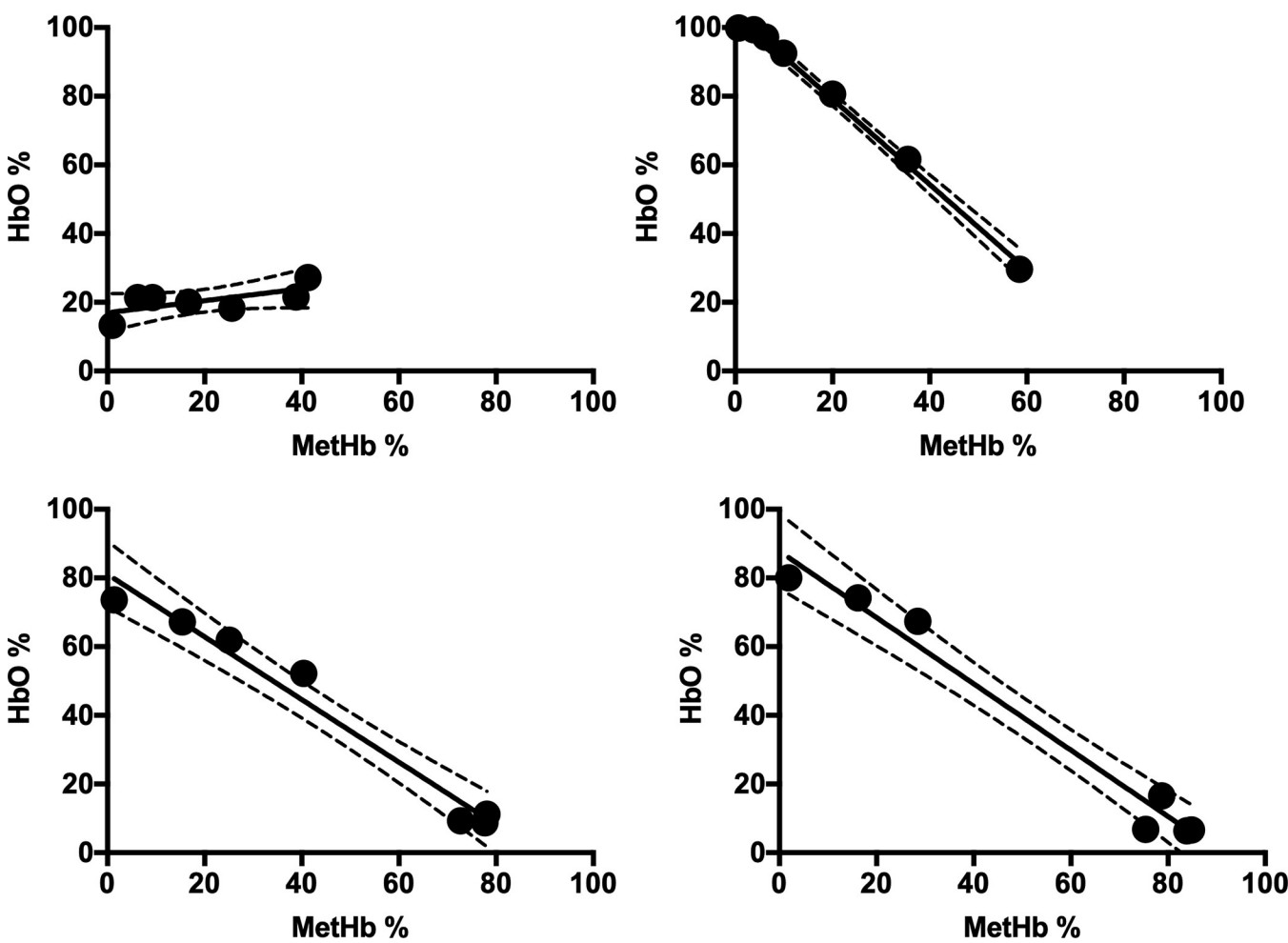

**Fig 4.** Regression of methaemoglobin (MetHb) and oxyhaemoglobin (HbO) determined by the Avoximeter CO-Oximeter in whole blood aliquots incubated of 0, 2.5, 5, 10, 15, 20, 40 mmol $L^{-1}$ $NaNO^2$ for (a) chicken, (b) possum, (c) cat and (d) dog blood.

changes to the absorbance values over the entire spectra were observed in day-old blood, although visual and spectroscopic examination found no obvious haemolysis present (Fig 5).

## Association of MAI with changing $FeHb^{2+}$ concentrations

Mean baseline MAI, LAC, Hbt and PCV were collected in 9 bird species prior to dosing with PAPP (Table 1). During *in vivo* comparisons in 9 bird species, blood samples typically analysed within 1 minute revealed prominent HbO absorbance peaks at 540 nm and 575 nm in all cases. As the 575 nm absorbance peak for HbO was found to be in the most linear relationship with a dose-dependent MetHb response at 630 nm each wavelength was adopted for the MAI.

In response to a wide range of oral PAPP doses, Cera-Check Hb($Fe^{2+}$) data and MAI absorbance spectrometer were found to be strongly and significantly correlated in all 9 species (Fig 6; Table 2).

## Discussion and conclusions

Together with LAC data from the Lactate Scout, our *in vivo* bird data were collected using 5 μL blood sample volumes. Requiring only a 1 μL blood sample, the Cera-Chek biosensor provided

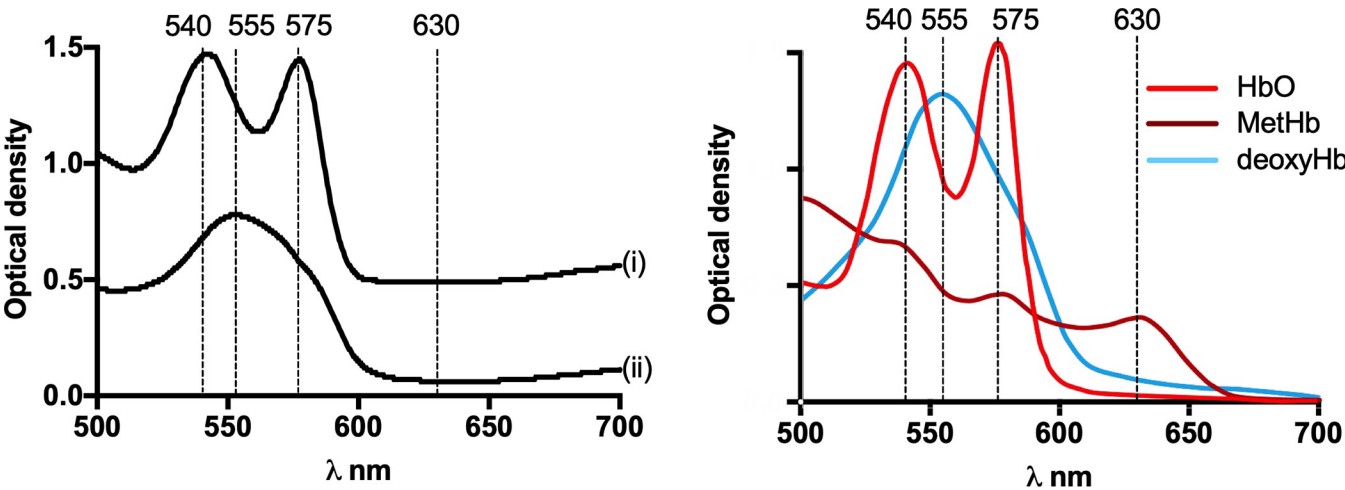

**Fig 5.** (a) Absorbance spectra (nm) of (i) fresh drawn chicken blood and (ii) the an aliquot preserved in lithium heparin at 6˚C for 24 hours in the laboratory. (b) Optical density of oxyhaemoglobin (HbO), methaemoglobin (MetHb) and deoxyhaemoglobin (deoxyHb) as reported by Van Kampen and Zijlstra (1966).

a useful relative indicator of declining $HbFe^{2+}$ in birds dosed with PAPP that closely corresponded with the declining MAI values, enabling a routine two-factor validation of data taken from all bird species. While the overall volume of blood sample wastage was difficult to quantify, even if twice the volume analysed was lost during collection, our procedures allowed us to harvest < 100 µL of blood in six sampling events, staying within our maximum blood volume target for a 10 g bird.

Our preliminary trials sought to validate instruments using different MetHb concentrations produced with a $NaNO_2$ gradient. These followed the methods described by (Shihana et al. 2010) for human erythrocytes that were validated in dog, cat and bird blood (Patton et al. 2016). However, as $NaNO_2$ concentrations > 60 mmol $L^{-1}$ caused varying degrees of hemolysis, we restricted our comparisons to an upper limit of 60 mmol $L^{-1}$. In chicken, possum, dog and cat blood *in vitro*, declining $Hb(Fe^{2+})$ values determined by the Cera-Chek biosensor were strongly correlated with MetHb concentration determined by the CO-Oximeter. In all three mammals, the strong inverse relationship between MetHb and HbO determined by the CO-Oximeter corresponded with *in vivo* data reported for carnivores [10] and reptiles [28] orally dosed with MetHb-forming agents. Our data also accorded with prior *in vitro* experiments using a similar $NaNO_2$ gradient with coyote, duck and starling erythrocytes that were

**Table 1. Mean T0 values for the methaemoglobinaemia absorbance index (MAI), lactate (LAC), haemoglobin ($HbFe^{2+}$) and packed cell volume (PCV).**

| Common name | Species | MAI | | | Lactate mmol l$^{-1}$ | | | HbFe$^{2+}$ g dl$^{-1}$ | | | PCV % | | |
|---|---|---|---|---|---|---|---|---|---|---|---|---|---|
| | | n | mean | SD | n | mean | SD | n | mean | SD | n | mean | SD |
| **Chicken** | *Gallus domesticus* | 9 | 0.977 | 0.174 | 9 | 1.04 | 0.38 | 9 | 13.72 | 0.99 | 9 | - | - |
| **Japanese quail** | *Coturnix japonica* | 15 | 1.614 | 0.141 | 15 | 2.65 | 1.03 | 15 | 21.09 | 1.47 | 15 | 63.4 | 4.3 |
| **Duck** | *Anas platyrhynchos* | 13 | 1.434 | 0.211 | 13 | 2.41 | 0.90 | 13 | 18.92 | 2.43 | 13 | 57.2 | 5.4 |
| **Pukeko** | *Porphyrio porphyrio* | 13 | 1.375 | 0.276 | 13 | 5.81 | 2.66 | 13 | 21.77 | 1.84 | 13 | 63.2 | 2.3 |
| **Black-backed gull** | *Larus dominicanus* | 8 | 1.639 | 0.079 | 8 | 2.3 | 2.63 | 8 | 23.48 | 1.15 | 8 | 70.5 | 3.4 |
| **Takahē** | *Porphyrio hochstetteri* | 10 | 1.403 | 0.162 | 10 | 4.25 | 1.41 | 10 | 20.98 | 1.62 | 10 | 62.7 | 4.6 |
| **Rosella** | *Platycercus eximius* | 10 | 1.472 | 0.140 | 10 | 1.67 | 0.85 | 10 | 19.53 | 1.32 | 10 | 58.8 | 4.0 |
| **Weka** | *Gallirallus australis* | 6 | 1.22 | 0.184 | 6 | 7.34 | 2.98 | 6 | 18.87 | 3.25 | 6 | 56.7 | 9.6 |
| **Brown kiwi** | *Apteryx mantelli* | 11 | 1.23 | 0.186 | 11 | 3.98 | 2.12 | 11 | 18.63 | 2.07 | 11 | 56.6 | 6.0 |

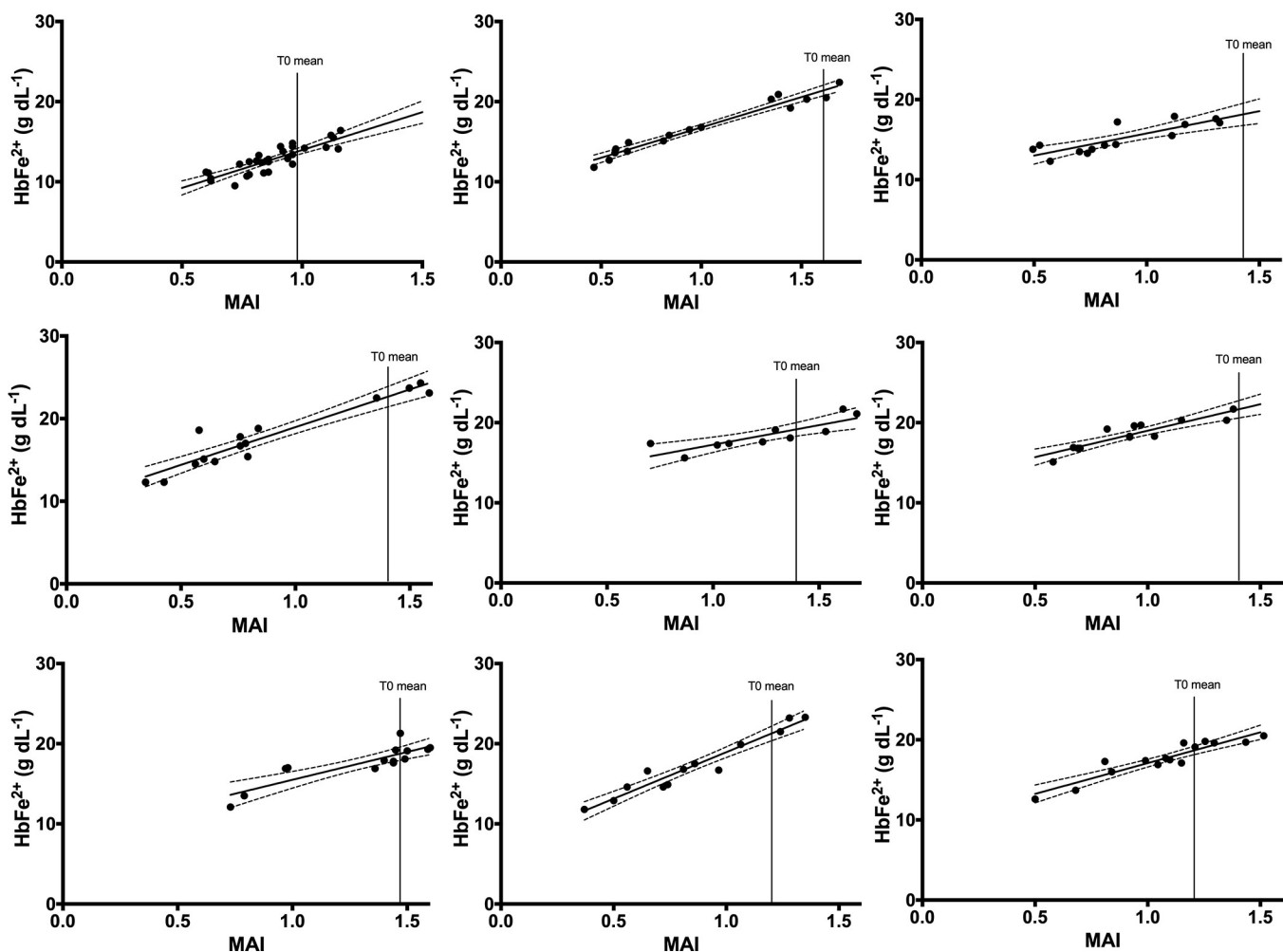

**Fig 6. Regression of absorbance data (MAI Eq 1 = 575 nm– 630 nm) with haemoglobin (HbFe$^{2+}$) at T1 (60 min) and T2 (180 min) for all birds dosed with PAPP (λ) at T0 with ST$_{99}$ (0.66), LT$_{50}$ (0.53) and T0 mean with 95% confidence.** Table 2 contains corresponding statistical information (a. Chicken; b. Japanese quail; c. Duck; d. Pukeko; e. Black-Backed Gull; f. Takahē; g. Eastern Rosella; h. Weka and i. Kiwi).

**Table 2. Regression of methaemoglobin absorbance index (MAI) with haemoglobin (HbFe$^{2+}$ g dL$^{-1}$) in the ferrous (Fe$^{2+}$) state for individual birds (n) dosed with PAPP at T0 and blood sampled at 60 and 180 minutes with the degrees of freedom (d.f.), R-squared (R$^2$), F-value (F) and significance of the regression fit (P).**

| Common name | Equation | n | d.f. | R$^2$ | F | P |
|---|---|---|---|---|---|---|
| Chicken | y = 9.457x + 4.502 | 6 | 1, 28 | 0.75 | 80.8 | < 0.0001 |
| Japanese quail | y = 7.57x + 9.211 | 9 | 1, 14 | 0.96 | 324.0 | <0.0001 |
| Duck | y = 5.545x + 10.24 | 7 | 1,12 | 0.7 | 28.1 | <0.0002 |
| Pukeko | y = 9.131x + 9.834 | 7 | 1, 13 | 0.90 | 113.4 | < 0.0001 |
| Black-backed gull | y = 4.89x + 12.36 | 5 | 1, 8 | 0.72 | 20.9 | < 0.002 |
| Takahē | y = 6.6x + 12.4 | 7 | 1, 11 | 0.82 | 50.5 | < 0.0001 |
| Rosella | y = 6.946x + 8.552 | 7 | 1, 12 | 0.75 | 36.5 | < 0.0001 |
| Weka | y = 11.62x + 7.324 | 6 | 1, 11 | 0.94 | 162.6 | < 0.0001 |
| Brown kiwi | y = 7.685x + 9.404 | 9 | 1, 13 | 0.87 | 87.9 | < 0.0001 |

lysed prior to spectroscopic and colorimetric determination of MetHb (Patton et al. 2016). In both cases, carnivores appeared to be substantially more prone to MetHb formation from direct $NaNO_2$ oxidation than bird blood.

Unlike mammalian blood, substantial *in vitro* sample degradation in chicken blood was observed within 24 hours, denoted by the loss of HbO absorbance peaks and increases in deoxyHb compared to freshly drawn samples. This is consistent with observations where avian erthyrocytes consume oxygen some 7–10 times faster than mammalian blood [34]. Although no hemolysis was detected in our day-old blood sample, widespread changes to absorbance values over the full spectra were observed, along with frequent error messages from the CO-oximeter when day-old blood was used. Chicken blood may be subject to rapid coagulation and is difficult to store even for short periods [35]. Anticoagulants used for the storage of nucleated erythrocytes are also associated with changes in blood values and hemolysis [36]. Because MetHb-forming agents cause hypoxia by diminishing the capacity for HbO formation and $O_2$ availability for aerobic metabolism, analysis using degraded bird blood risks producing data that do not accord with the true pathological response to MetHb-forming agents *in vivo*. Our results agree with other studies that concluded that analysis of non-mammalian blood immediately after collection is warranted and justified by the unique physiology of mammalian erythrocytes compared to other vertebrates. In contrast to all other classes of vertebrates (reptiles, fish, birds and amphibians), mammalian erythrocytes lose their nucleus and mitochondria during erythroblast maturation [37] and are thereafter entirely dependent upon anaerobic (fermentation) glycolysis to generate ATP and reducing equivalents [38]. In contrast, bird erythrocytes contain active mitochondria that consume $O_2$ during oxidative metabolism [39] in addition to transporting $O_2$ as HbO. Unlike mammals, MetHb-reductase activity in bird erythrocytes may also be oxidative [29] and its rate determined by energy flux reliant upon $O_2$ remaining in a blood sample over time. Metabolic demand for $O_2$ in avian erythrocytes and their dependence upon a supply of Tricarboxylic acid (TCA) cycle energy substrates to support oxidative metabolism [40] may partly account for why bird blood has been observed to deteriorate almost immediately after collection [35]. Studies of MetHb formation in stored blood also reveal that complex interactions involving ROS and strongly oxidising ferryl ($HbFe^{4+}$) compounds cycling between the ferryl, ferric and ferrous state, interacting with a range of redox enzymes [41]. Because oxidative stress caused by severe methaemogobinaemia may lead to haemolysis and the dissociation of the Hb tetramer [42] and release of free $Fe^{2+}$, this may give rise to hydroxide ion and hydroxyl free radicals via the Fenton reaction [43] that produce further MetHb after sample collection. Accordingly, analysis of whole blood samples at the point of sample collection minimises possible analytical and storage artifacts arising from sample degradation that appears to be much less of a problem in mammalian erythrocytes. However, if sulphaemoglobin (SHb) and carboxyhaemoglobin (HbCO) are present within the blood sample, the determination of MetHb and HbO by absorbance spectroscopy can be confounded under certain circumstances. Combined, SHb and HbCO are normally present at only 1–2% [44] and are unlikely to interfere substantially with the MAI unless SHb is formed in far greater concentrations due to the presence of specific xenobiotic compounds and the action of some bacteria [45]. Practically, only in the presence of significant atmospheric concentrations of carbon monoxide gas will HbCO levels become elevated enough to interfere in the determination of HbO and influence the MAI.

Methaemoglobinaemia, is a functional anaemia where MetHb displaces HbO, in a relationship that is strongly inversely correlated [10, 28]. Individually, neither MetHb or HbO concentrations proved to be as successful at predicting bird survival as the MAI [17]. In small blood samples, relative changes in absorbance between 575 nm and 630 nm appeared far less affected by error than those referenced to zero baselines that might shift slightly between samples due

to imperfections in the cuvette repositioning or changes in the optical characteristics of the sample itself. Although the MAI was in most cases strongly correlated with declining $HbFe^{2+}$ concentrations, several outliers weakened the coefficient of determination in some species. While point-of-care devices have proven highly useful in gathering haematology data in wildlife species, the cause of variation and error should ideally be qualified and correction factors developed if appropriate [46]. Given our priority to analyse bird blood samples within a 1-minute window after collection we did not obtain data to test whether these outliers were associated due to poor repeatability or differences in sampling times. However, we noted that changes in blood absorbance values within the 3-µL cuvette often remained dynamic after analysis. Slight shifts at both 575 nm and 630 nm were observable in the real-time on the spectrometer display after data had been recorded that suggested that oxidation of haemoglobin continued within the sample, most likely due to the intraerythrocytic catalytic cycle where the PAPP active (para-hydroxylaminopropriophenone) is regenerated, accounting for its slow rate of elimination and high potency [47, 48]. A failure to closely synchronise the collection of spectral and $HbFe^{2+}$ data arose on several occasions, mostly due to errors in the filling of cuvettes that occasionally necessitated their reloading and replacement. Closer temporal alignment of data collection or replication of Cera-Check $HbFe^{2+}$ data using an average of multiple readings, may assist to reduce outliers. Discrepancies between absorbance-based measurements of haemoglobins and those using biosensors reliant on different analytical principles may also occur due to haemolysis [49], or if the concentration of reticulocytes varies between individuals [50], neither of which we sought to qualify. Unlike laser flow cytometry used in automatic haematology analysers, we obtained no information about the size and shape of cells [51], meaning that it is also possible that variation in several erythrocyte indices, such as mean crepuscular volume (MCV) could also have accounted for variations in absorbance data relative to total $HbFe^{2+}$ concentrations [52]. Potentially the integration of validated PCV data, to produce the mean corpuscular haemoglobin concentration (MCHC), may prove to be a more robust contrast with our absorbance data.

## Acknowledgments

The authors wish to thank the Manaaki Whenua Landcare Research (MWLR) Animal facility staff for husbandry of domestic and wild birds, and MWLR field staff for capture of wild birds; Kate McInnes and Lydia Uddstrom (DOC veterinarians) for helping develop and refine the methodology for native species; The Takahē team, particularly the staff and vet at the Burwood Takahē Centre and the staff at the Dunedin Wildlife Hospital for their invaluable assistance; Te Kaitiaki Rōpū O Murihiku (Kākāpō/ Takahē Iwi advisory board) for supporting this work; The Otorohanga Kiwi House management, staff and attending vet, for their commitment and assistance; and Ngāti Hinewai kaumatua for supporting the research; The Kiwi and Weka recovery groups (DOC); and the NZ Zoo and Aquarium Association for facilitating contact with the captive facilities. Dr Elaine Murphy's inputs and feedback on the original research proposal for this work and ongoing data collection were significant and greatly appreciated. We thank Dr Penny Fisher for coordinating the original NWR research proposal with MWLR staff and DOC and providing comments on an early progress report. The paper benefited from the constructive feedback from two anonymous referees and Professor Stuart McLean (University of Tasmania). The authors would like to acknowledge the manu used in this research. We hope their species thrive.

## Author Contributions

**Conceptualization:** Clive A. Marks.

**Data curation:** Katherine Trought, Samantha Brown, Jane Arrow.

**Formal analysis:** Clive A. Marks.

**Funding acquisition:** Clive A. Marks, Brian Hopkins.

**Investigation:** Clive A. Marks, Katherine Trought, Samantha Brown, Jane Arrow.

**Methodology:** Clive A. Marks, Katherine Trought.

**Project administration:** Clive A. Marks, Samantha Brown, Brian Hopkins.

**Supervision:** Brian Hopkins.

**Validation:** Clive A. Marks, Samantha Brown.

**Visualization:** Clive A. Marks.

**Writing – original draft:** Clive A. Marks.

**Writing – review & editing:** Clive A. Marks, Katherine Trought, Samantha Brown, Jane Arrow, Brian Hopkins.

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
