## [Decision Letter · Decision Letter 0]

5 Jan 2023

PONE-D-22-31643Monitoring methaemoglobinaemia in birds using 5 µL of whole bloodPLOS ONE

Dear Dr. Marks,

Thank you for submitting your manuscript to PLOS ONE. After careful consideration, we feel that it has merit but does not fully meet PLOS ONE’s publication criteria as it currently stands. Therefore, we invite you to submit a revised version of the manuscript that addresses the points raised during the review process.

We look forward to receiving your revised manuscript.

Kind regards,

Robert Chapman, Ph.D.

Academic Editor

PLOS ONE

Journal Requirements:

3. To comply with PLOS ONE submissions requirements, in your Methods section, please provide additional information regarding the experiments involving animals and ensure you have included details on (1) methods of sacrifice, (2) methods of anesthesia and/or analgesia, and (3) efforts to alleviate suffering.

"This work was funded by the Department of Conservation (New Zealand) Predator Free 2050 Tools to Market Programme under contract (Reference Number: 3054988; Contract Report: LC4147)"

7. We note that Figure 1 in your submission contain copyrighted images. All PLOS content is published under the Creative Commons Attribution License (CC BY 4.0), which means that the manuscript, images, and Supporting Information files will be freely available online, and any third party is permitted to access, download, copy, distribute, and use these materials in any way, even commercially, with proper attribution. For more information, see our copyright guidelines: http://journals.plos.org/plosone/s/licenses-and-copyright.

Additional Editor Comments:

This paper is a well written and detailed study that is rigorous in its analysis. Like reviewer 3, I do not understand why the analysis has been performed using MAI, and this could be clarified by minor revision. Line 232 is confusing in this respect: "As the 575 nm absorbance peak for HbO was found to be in the most linear relationship with a dose-dependent MetHb response at 630 nm each wavelength was adopted for the MAI."

Reviewers' comments:

Reviewer's Responses to Questions

**Comments to the Author**

1. Is the manuscript technically sound, and do the data support the conclusions?

Reviewer #1: Yes

Reviewer #2: No

Reviewer #3: Partly

2. Has the statistical analysis been performed appropriately and rigorously? 

Reviewer #1: Yes

Reviewer #2: No

Reviewer #3: I Don't Know

3. Have the authors made all data underlying the findings in their manuscript fully available?

Reviewer #1: Yes

Reviewer #2: Yes

Reviewer #3: No

4. Is the manuscript presented in an intelligible fashion and written in standard English?

Reviewer #1: Yes

Reviewer #2: No

Reviewer #3: Yes

5. Review Comments to the Author

Reviewer #1: This is a well-conceived study that has achieved its objective, to find a method of determining the level of methaemoglobinaemia in the small volume of blood that can be taken from small birds. This is needed for toxicological studies of the effects of baiting with a methaemoglobin-inducing agent (PAPP) on non-target species, mainly birds. The new method demonstrates how a selection of commercially-available devices can be combined with bespoke equipment to create a method not otherwise available. The study is well-conducted, the data analysis is sound, the findings are clearly presented and the conclusions are justified.

Figures

There are very many figures that show similar data. Perhaps some could be placed in a Supplementary section. This could make the narrative flow more easily.

Minor issues

Line no.

18, total Hb is not HbFe2+, but needs definition here – later called Hbt (line 203)

44 word order?

80-81 re-order words?

142- birds were given increasing doses of PAPP until a clinical response. The actual doses used varied between species and should be indicated.

203-6 give N animals used for each correlation analysis

298- in this discussion of sources of variation in MAI, has consideration been given to the possibility of further oxidation products of Hb, such a sulf-Hb?

Reviewer #2: The concept of measuring metHb concentration in small blood volumes has merit. However, this manuscript does not explain the data in a way that is useful and can be understood by others significantly familiar with the field. There are fundamental problems with the way the MAI is calculated in that the peaks at 575 nm will never go to zero and the contribution of the MetHb to the peak at 575 needs to be accounted for in all measurements. If the authors plan on submitting this manuscript again, it needs to be redone to improve the clarity of sentences.

Reviewer #3: This reviewer appreciates the care taken by the authors to develop methods that mitigate risks and death to study animals. Furthermore, obtaining point-of-collection data via spectroscopy on small sample volumes is very difficult, and this reviewer appreciates the achievements reported in the manuscript.

This reviewer recommends several modifications to the manuscript to clarify the presentation and interpretation of results prior to publication. Specific areas to be addressed are given below with line and figure references.

Line 34: The statement “…methaemoglobin (MetHb), it is rendered unable to release O2 from the tetrameric Hb molecule…” is misleading since MetHb is unable to bind and carry O2 (Mansouri and Lurie, 1993, American Journal of Hematology). This reviewer suggests modifying the text to clarify this point.

Lines 136-138: It is unclear what volume of NaNO2 in PBS was added to blood samples to induce methemoglobin formation and whether the cells are lysed by the addition of this solution. Please add text to clarify and comment on whether cell lysis and subsequent reduction in light scattering has an impact on the spectroscopic analysis with the custom apparatus.

Lines 184-186: It is unclear why the authors chose to report results in terms of a Methemoglobin Absorption Index (MAI) when the change in absorbance at 630nm (MetHb) should return the same information. The biggest concern with this is the convolution of the pure HbO spectrum with the pure MetHb spectrum, so while HbO is being depleted, MetHb is growing in at the chosen wavelength of 575nm. Will the authors please comment on this chosen reporting method and clarify in the text?

Line 187: It is concerning that there is a nonzero baseline for Figure 2. What is the origin of this and is the baseline consistent across all measurements? A nonzero baseline is typically an indication of scatter either from the sample or the sample holder. If scattering in the sample is dynamic and changing over the course of the experiment, this would affect the linearity of the MAI plot (Figure 2b). If it changes from one experiment to the next it could shift the MAI up or down with respect to the y-axis. This effect would be most pronounced in experiments where samples are subject to increased cell lysis due to higher ionic strength (increasing nitrite concentrations). This would impact the interpretation of data presented in figures 3 and 4.

Figures 3 & 4 and Lines 195-197: Please clarify what is meant by “…all MAI data… were regressed against Hbt…”. What statistical treatment is being done? Or are the data simply plotted together on the same graph? Also, please clarify which data is indicated by the markers, solid line, and dashed lines. It seems that the authors are plotting (Hbt vs. NaNO2) data with (MetHb vs. NaNO2) data, but NaNO2 concentrations are not shown in the plot.

Figure 5: Again, a nonzero baseline for the HbO spectrum (figure 5a) this time in freshly drawn chicken blood indicates scattering due to sample or sample holder. A change in OD at 575nm could be due to a reduction in scattering from cell lysis or sedimentation. This would complicate the interpretation of results reported as a change in HbO percentage with time and treatment. Additionally, the absence of tick labels for figure 5b makes it impossible to determine whether scattering is occurring in the sample.

6. PLOS authors have the option to publish the peer review history of their article (what does this mean?). If published, this will include your full peer review and any attached files.

Reviewer #1: **Yes: **Stuart R McLean.

Reviewer #2: No

Reviewer #3: No

---

## [Author Response · Author response to Decision Letter 0]

2 Feb 2023

All of these responses are in the attached documents that have been uploaded - cover letter and response to reviewers document. Why is the system asking for this to be pasted here when all of the formatting is lost? Please advise.

---

## [Editor Report · Decision Letter 1]

24 Feb 2023

Monitoring methaemoglobinaemia in birds using 5 µL of whole blood

PONE-D-22-31643R1

Dear Dr. Marks,

We’re pleased to inform you that your manuscript has been judged scientifically suitable for publication and will be formally accepted for publication once it meets all outstanding technical requirements.

Kind regards,

Robert Chapman, Ph.D.

Academic Editor

PLOS ONE

Additional Editor Comments (optional):

Thank you for the detailed response to all of the concerns and questions raised by the reviewers. There is no doubt that this manuscript meets PLOSone's publication criteria and I enjoyed reading through the discussion. The key point that required clarification for the purposes of meeting PLOSone's criteria is the one raised by reviewer #2 - whether or not it is valid to measure the MAI in this way. It is clear to me that your approach is correct - certainly the overlapping spectra of MetHB and HbO does not cause a problem.. There will be some degree of uncertainty in the obtained values, but the use of this method as a predictive tool of survival in other literature is convincing. Your response to the other comments raised by reviewer #3 clarifies the issues, and the changes made in the manuscript seem very sensible to me.

---

## [Editor Report · Acceptance letter]

7 Mar 2023

PONE-D-22-31643R1 

Monitoring methaemoglobinaemia in birds using 5 µL of whole blood 

Dear Dr. Marks:

I'm pleased to inform you that your manuscript has been deemed suitable for publication in PLOS ONE. Congratulations! Your manuscript is now with our production department. 

Kind regards, 

on behalf of

Dr. Robert Chapman 

Academic Editor

PLOS ONE